

# The anatomy of the foveola reinvestigated

Alexander V. Tschulakow[1], Theo Oltrup[2], Thomas Bende[2], Sebastian Schmelzle[3] and Ulrich Schraermeyer[1,4]

[1] Division of Experimental Vitreoretinal Surgery, Centre for Ophthalmology, University Hospital Tübingen, Tübingen, Germany
[2] Division of Experimental Ophthalmic Surgery, Centre for Ophthalmology, University Hospital Tübingen, Tübingen, Germany
[3] Ecological Networks, Department of Biology, Technische Universität Darmstadt, Darmstadt, Germany
[4] Ocutox (www.ocutox.com), Hechingen, Germany

Corresponding author
Ulrich Schraermeyer,
u.schraermeyer@gmail.com,
Ulrich.Schraermeyer@med.uni-tuebingen.de

## ABSTRACT

**Objective**. In the foveola of the eye, photoreceptors and Müller cells with a unique morphology have been described, but little is known about their 3D structure and orientation. Considering that there is an angle-dependent change in the foveolar photoreceptor response for the same light beam, known as the Stiles Crawford Effect of the first kind (SCE I), which is still not fully understood, a detailed analysis of the anatomy of the foveolar cells might help to clarify this phenomenon.

**Methods**. Serial semithin and ultrathin sections, and focused ion beam (FIB) tomography were prepared from 32 foveolae from monkeys (*Macaca fascicularis*) and humans. Foveolae were also analyzed under the electron microscope. Serial sections and FIB analysis were then used to construct 3D models of central Müller and photoreceptor cells. In addition, we measured the transmission of collimated light under the light microscope at different angles after it had passed through human foveae from flat mounted isolated retinae.

**Results**. In monkeys, outer segments of central foveolar cones are twice as long as those from parafoveal cones and do not run completely parallel to the incident light. Unique Müller cells are present in the central foveolae (area of 200 μm in diameter) of humans and monkeys. Light entering the fovea center, which is composed only of cones and Müller cells, at an angle of 0° causes a very bright spot after passing through this area. However, when the angle of the light beam is changed to 10°, less light is measured after transpasssing through the retina, the foveolar center becomes darker and the SCE-like phenomenon is directly visible. Measurements of the intensities of light transmission through the central foveola for the incident angles 0 and 10° resemble the relative luminance efficiency for narrow light bundles as a function of the location where the beam enters the pupil as reported by Stiles and Crawford. The effect persisted after carefully brushing away the outer segments.

**Conclusion**. We show that unique cones and Müller cells with light fibre-like properties are present in the center of the fovea. These unique Müller cells cause an angle dependent, SCE-like drop in the intensity of light guided through the foveola. Outer segments from the foveolar cones of monkeys are not straight.

## INTRODUCTION

The primate retina contains two types of photoreceptors, rods for night vision and cones for daylight vision. Cones are predominately located in the macula lutea. This has a diameter of around 5.5 mm in humans and is subdivided into the fovea and parafovea with diameters of 1.8 mm and 2.3 mm, respectively (*Hogan, Alvarado & Weddell, 1971*).

The fovea is a small pit in the retina which contains the largest concentration of cones and is responsible for sharp central vision. The central part of the fovea is called the foveola (*Hogan, Alvarado & Weddell, 1971*) and has been regarded as being 350 µm in diameter since 1941 (*Polyak, 1941*). Only in the foveola does visual acuity reach 100 percent (*Trauzettel-Klosinski, 2010*).

The original discovery by Stiles and Crawford described that the apparent brightness of an object is not proportional to the pupil area because light rays entering the pupil distant from the axis are not so visually effective as rays entering along or near to the central axis (*Stiles & Crawford, 1933*).

Stiles also showed that monochromatic light will differ in hue between on- and off-axis even after the light beams are equated for brightness. This was described as the Stiles–Crawford effect of the second kind (SCE 2).

The Stiles Crawford Effect of the first kind (SCE I) was regarded as one of the most important discoveries in visual science of the last century (*Westheimer, 2008*). The explanation of the SCE has so far been handled in abstract models involving specific photoreceptor orientation, subtle morphological differences in rods and cones, anchoring of photo-pigment molecules in membranes or possible phototropism of retinal cells (*Laties, 1969*). But none of these models could be proven and a full explanation of the SCE continues to provide challenges (*Westheimer, 2008*).

Since the first report of Stiles and Crawford, a large body of histological and psychophysical evidence has accumulated (for review see *Westheimer, 2008*) showing that cones in different retinal regions are directionally sensitive.

It was speculated that a change in the shape or the orientation of foveal cones was probably responsible for the SCE (*Westheimer, 1967*). But until now, no morphologic evidence for this assumption has been found, and in contrast a different orientation of foveal and parafoveal cones in monkeys and humans (*Hogan, Alvarado & Weddell, 1971*) was ruled out by histologic examinations more than four decades ago (*Laties, 1969*; *Westheimer, 2008*). To study the SCE I, monkey eyes are suitable because human and monkey foveae are very similar (*Krebs & Krebs, 1991*) and the existence of the SCE I has also been demonstrated for monkeys (*Matsumoto et al., 2012*).

The most renowned publication about the histology of the human eye (*Hogan, Alvarado & Weddell, 1971*) states that in meridional sections of the foveolar region, the cones are perfectly straight and oriented vertically with respect to the retinal surface, and their axes are parallel to each other. Thus, the view that foveal cones lack directional morphology has remained valid right up to the present day.

Clinically the SCE I is used for diagnosis of macular telangiectasia type 2 (*Charbel Issa et al., 2016*) which is a poorly understood condition of the retina that may result in blindness (*Charbel Issa et al., 2013*).

The SCE is absent in rod photoreceptors (*Lu et al., 2013*) leading to the speculation that differences in photo-pigment structure and anchoring of cones and rods may be involved in SCE (*Enoch & Stiles, 1961*; *Walraven & Bouman, 1960*; *Westheimer, 2008*). Despite its pivotal role for sharp central vision the definite anatomy of the fovea at high resolution is not known (*Yamada, 1969*).

The aim of our study was to investigate the 3D anatomy of foveolar cells (Fig. 1) and whether it can help to explain the SCE I.

## MATERIALS AND METHODS

### Light and electron microscopy from monkey eyes

Twenty-four monkey eyes (*Macaca fascicularis*, 14 males, 10 females) were collected after sacrificing the animals under general anesthesia, i.e., intramuscular injection of ketamine hydrochloride followed by an intravenous sodium pentobarbitone (Lethabarb®, Virbac, Australia) overdose. Monkeys were kept at Covance Laboratories GmbH (Münster, Germany study numbers 0382055, 8260977, 8274007) or SILABE-ADUEIS (Niederhausbergen, France). The Covance Laboratories GmbH test facility is fully accredited by the Association for Assessment and Accreditation of Laboratory Animal Care (AAALAC). This study was approved by the local Institutional Animal Care and Use Committee (IACUC), headed by Dr. Jörg Luft and the work was carried out in accordance with the Code of Ethics of the World Medical Association (Declaration of Helsinki). The monkeys from SILABE-ADUEIS were euthanized due to veterinarian reasons. Since they had not been included in a study before, they do not have a study number. The age of the monkeys varied between four to eight years. The eyes were enucleated 5 min post-mortem, cleaned of orbital tissue, and were slit carefully at the limbus without damaging the ora serata. Then, 200 µl of the fixative (5% glutaraldehyde) were carefully injected into the center of the vitreous. The intraocular pressure was balanced because vitreous could leak out from the opening thereby compensating the volume of the fixative. This protocol has been shown to minimize fixation artefacts according to our own experience. The eyes were then fixed at 4 °C by immersion into 5% glutaraldehyde in 0.1 M cacodylate buffer (pH 7.4; Sigma, St. Louis, MO, USA) overnight for electron microscopy. Glutaraldehyde fixed specimens were post-fixed with 1% $OsO_4$ at room temperature in 0.1 M cacodylate buffer (pH 7.4), stained with uranyl acetate, and the maculae were excised and embedded in Epon after dehydration in a graded series of ethanol and propylene oxide. Semi-thin sections were stained with toluidine blue and examined by light microscopy (Zeiss Axioplan 2 imaging; Zeiss, Jena, Germany). For electron microscopy, ultrathin sections were made and analyzed with a Zeiss 900 electron microscope (Zeiss, Jena, Germany). The foveae from 24 eyes were sectioned in a sagittal plane until the center of 21 foveolae was found. In three eyes, the foveal centers were missed. The center of the foveola was defined as the site where the cell fiber layers at the bottom of the foveal pit were free of nuclei and were 10 µm thin or less.

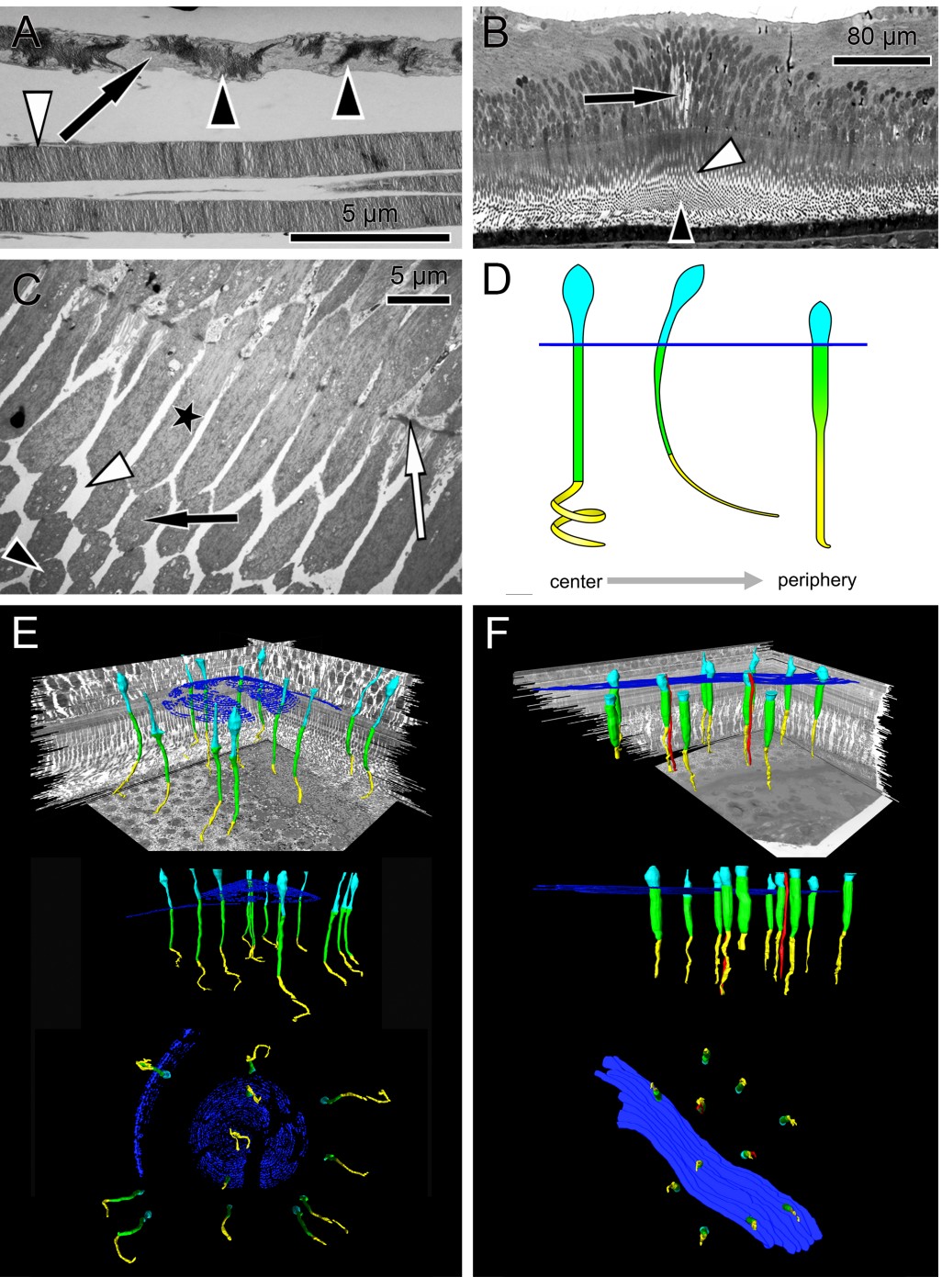

**Figure 1  Anatomical findings in monkey foveae and cones.** (A) In an electron micrograph, an extra foveal cone contains irregularly ordered stacks of photoreceptor disc membranes (black arrowheads) and spaces free of photoreceptor membranes (arrow). In contrast, the disk membranes in rods are highly ordered (white arrowhead). (B) In a semi-thin transverse section of the central fovea, the prominent Müller cells are present (arrow). Inner segments are curved and are (continued on next page...)

**Figure 1 (…continued)**
cut transversely in the plane of section. Spaces between the curved inner segments form symmetrical patterns (white arrowhead) (see also C and the model in D). Also in the middle of the subretinal space the outer segments are cut transversely (black arrowhead). (C) In an electron micrograph sectioned parallel to the optical axis through the central fovea, the inner segments of the cones are cut longitudinally (asterisk), diagonally (black arrow) and transversely (black arrowhead) in the same plane of section indicating their curved nature. The spaces between the curved inner segments shown in (B) are marked (white arrowhead). The white arrow marks the outer limiting membrane. (D) Different shapes of cones are presented schematically from the central fovea to the parafovea. (E + F) Central cones (E) and parafoveal cones (F) are shown in different views integrated into the retinal environment (top), from the front (middle) or from the RPE towards the vitreous (bottom). Parafoveal rods are marked in red, inner segments in green, outer segments in yellow and the outer limiting membrane in blue.

### Light and electron microscopy from human eyes

Three human eyes from a 57 and 81-year-old male, and a 68-year-old female were fixed 8 and 12 h postmortem at 4 °C by immersion into 5% glutaraldehyde in 0.1 M cacodylate buffer (pH 7.4; Sigma, St. Louis, MO, USA) for half an hour. Then the cornea was removed and fixation continued overnight for electron microscopy. The human eyes were gifts from the Clinical Anatomy of the University of Tuebingen (ethical number for scientific issues 237/2007B01), and taken from full body donors, who had previously given informed consent. Embedding and sectioning was done as described for the monkey eyes.

### Evaluation of serial sections through the fovea of monkey and humans

For 3D model reconstruction, semi-thin (700 nm) serial sections were performed from 6 monkey foveae.

Series 1 and 2 comprised 21 sections which correspond after mounting to a foveal tissue piece with a thickness of 14.7 μm. Unexpectedly, due to their curved shape, a foveal cone did not completely fit into such a tissue block. Therefore series 3 and 4 were sectioned in sagittal planes with 41 and 160 sections respectively. Additionally, series 5 and 6 were cut as transverse sections with 450 and 195 sections respectively. Sagittal sections run within or parallel to the optical axis, while transverse sections were made perpendicular to it.

Serial transverse sections were performed from two human foveae. Series 7 from a 68-year-old female and series 8 from an 81-year-old male contained 250 and 460 sections, respectively. From each section, the fovea was photographed at a 600-fold magnification.

### 3D modelling

The 3D reconstructions of the presented figures and measurements were performed with Amira® software (version 5.6; FEI, Hillsboro, OR, USA). Using previous embedded fixed marker, the images of the sections were aligned manually following digitization by comparing superimposed slices, translating, and rotating adjacent slides with respect to one another. Additionally, the border of each slide and the regular structures and patterns were used as markers for alignment.

Once the aligned sections were imported into Amira, the structures of interest were labeled using the software segmentation tools. The structures of interest were cone nuclei, inner and outer segments, the outer limiting membrane, and Müller cells. The length of the

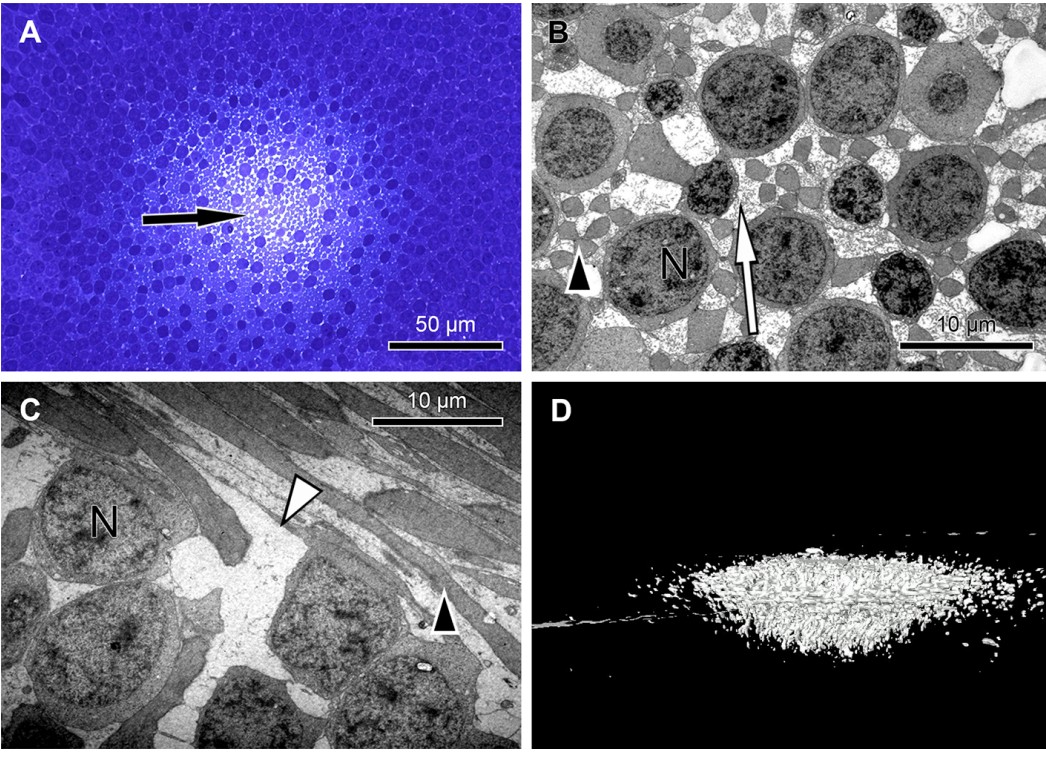

**Figure 2** **Unique central foveolar Müller cells from monkeys.** In the fovea there are only two types of cells. Müller cells appear white or electron-lucent and cone photoreceptors appear blue or electron-opaque. Thus they can be easily distinguished. (A) In a semi-thin section perpendicular to the optical axis the central Müller cells are translucent (arrow). (B) The Müller cells (arrow) are shown in the same orientation as in (A) at high magnification. They do not contain cell organelles at this topographic site and appear white or electron-lucent. The arrowhead marks a Henle fibre. (C) The plateau zone of a Müller cell is indicated by a white arrowhead in a section parallel to the optic axis. Henle fibers are indicated by a black arrowhead and cone nuclei by (N). (D) A 3D model shows the main part of the central Müller cells of a monkey.

inner and outer segments of cones from the parafovea and the central fovea were measured using the ''Amira-3D length—measurement-tool''. For Fig. 2D and Video S1, areas of interest were selected by visually adjusting the grey value threshold of the volumetric dataset.

## Focused ion beam/scanning electron microscopy

Focused ion beam/scanning electron microscopy (FIB/ SEM) tomography data were acquired using a Zeiss Auriga CrossBeam instrument at the Natural and Medical Sciences Institute at the University of Tuebingen (NMI, Reutlingen, Germany) as described (*Steinmann et al., 2013*). For FIB/SEM analysis, the block of the embedded sample was sputter coated with gold palladium and mounted on an appropriate SEM sample holder. A semi-thin section of the embedded sample was imaged with the light microscope and correlated with the SEM image of the ultramicrotome block face to define the region of interest for three-dimensional analysis. Using a Crossbeam instrument (Zeiss) equipped

with a gallium FIB and a low voltage SEM, FIB/SEM serial sectioning tomography was accomplished. The gallium FIB produces thereby a series of cross-sections containing the region of interest. Each of these cross-sections is imaged by the low keV SEM using the energy-selected backscattered (EsB) electron detector for image acquisition. The following parameters were used for SEM: Primary energy of 1.8 keV with the aperture 60 µm: for image acquisition the EsB detector was used with a grid voltage of −1,500 V, i.e., only backscattered electrons with a maximum energy loss of 300 V were used for image acquisition. The resolution was 2,048 × 1,535 pixels with a pixel size of 42.41 nm.

For FIB, the following parameters were used: Primary energy of 30 keV, slicing was performed with a probe current of 2 nA, the slice thickness was 42 nm. In this way cubic voxels were obtained, i.e., the same resolution in $x, y$, and $z$, which is good for the reconstruction. The resulting stack of two-dimensional images was utilized for three-dimensional reconstruction using appropriate software.

## Wholemount preparations from human retinae

The maculae of five eyes from three donors were excised using a trephine with 1 cm diameter. The donors were two females of 74 and 90 years old, and one male of age 18. Three maculae were fixed in formalin transferred into phosphate buffered saline and mounted on slides covered with a cover glass on wax feet to prevent squeezing of the retinae. Another two eyes were treated identically but fixation was omitted. Finally, the maculae were observed under the light microscope having the condenser replaced by a modification with adjustable mirror (Fig. 3C).

## Illumination optics for measuring angle dependent light transmission through Müller cells

The optical fiber homogenizes the light of the light-emitting diode (LED). The light cone at the end of the fibre is collimated with an aspherical lens. With the focal length $f_{col}$, the core diameter $D_{fiber}$ defines the divergence angle $\theta_{div}$ and the opening angle $\theta_{NA}$ defines the diameter $D_{spot}$ of the light spot on the sample. The mirror can be tilted and slid. The angle of incidence of light on the sample is $\alpha_{slide}$.

To homogenize the intensity distribution of the light spot to be projected, the light from a light emitting diode (high power white LED, type: OSLON SSL, LCW CP7P.PC) is coupled to an optical fibre with the core diameter $D_{core} = 200$ µm (Fig. 3C). Through the principle of total reflection, the incident light in this optical conductor is reflected on the wall several times. In this way, the light is mixed and emerges homogeneously from the conductor. The light cone emitted in the opening angle $\theta_{NA}$ is collimated by an aspheric lens of focal length $f_{col} = 35$ mm. Since the light is not a point source, the rays diverge slightly. The divergence angle is in paraxial approximation $\theta_{div} = D_{core}/f_{col} = 0.2$ mm/35 mm = 5.7 mrad (0,3°). The diameter $D_{spot}$ of the projected light spots is given as NA = 0.16 through the focal length $f_{col}$ of the lens and the numerical aperture of the optical fibre. With NA $= \theta_{NA}/2, D_{spot} = 2 \cdot f_{col} \cdot$ NA $= 2 \cdot 35$ mm $\cdot$ 0,16=11,2 mm. The light striking the slide at an angle $\alpha_{slides}$ ispartially reflected according to the Fresnel equations and should be taken into account when measuring the intensity on the microscope. In normal incidence, $\alpha_{slide} = 0°$,

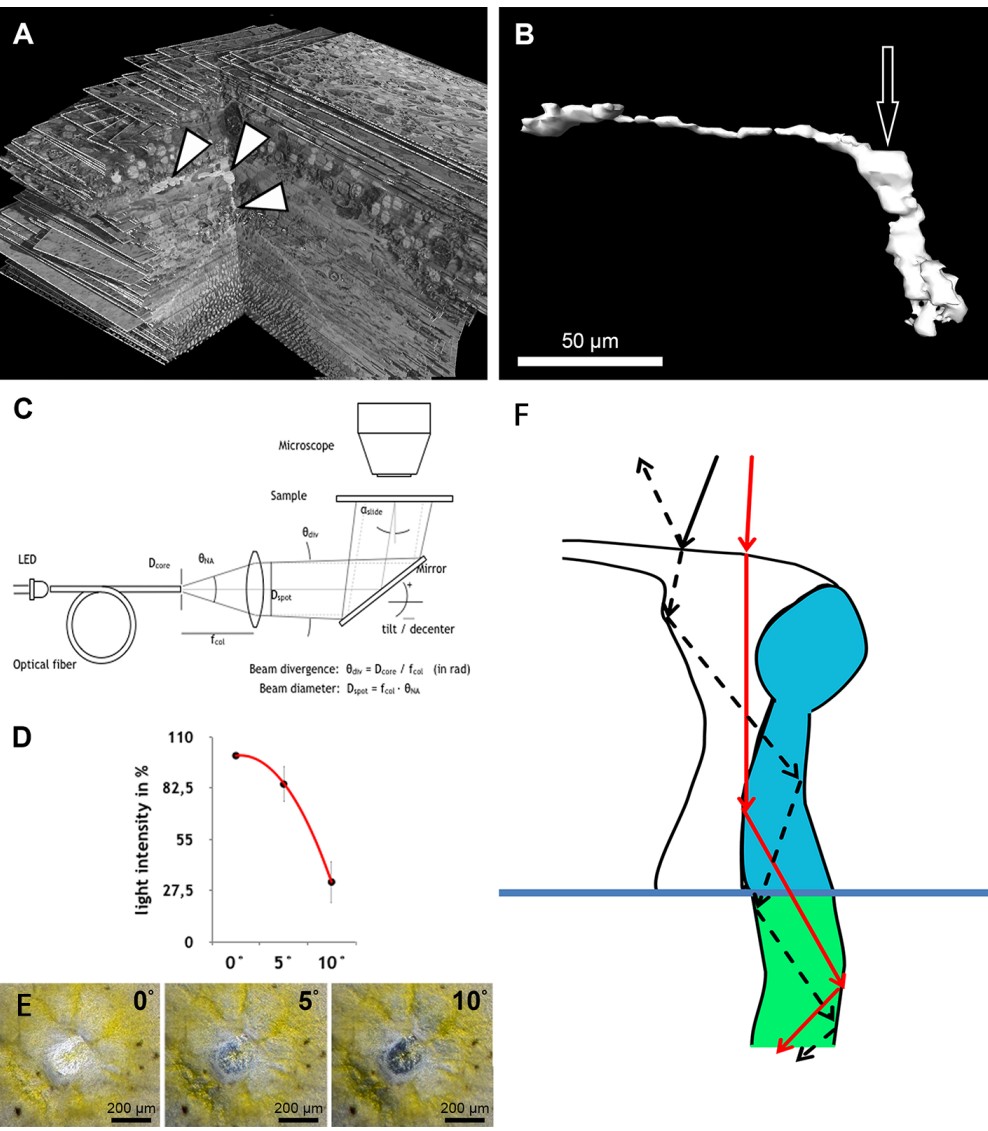

**Figure 3** **Measurement of light intensity after transpassing human foveolae.** (A) The central Müller cell (arrowheads) is shown integrated into a stack mounted from serial sections. (B) A human foveolar Müller cell 3D model is shown and its plateau zone marked by an arrow. (C) Schematic drawing of the measurement equipment is shown. Illumination optics for angle measurement at the Müller cells. The optical fiber homogenizes the emitted light of the light-emitting diode (LED). The light cone at the fiber end is collimated with an aspherical lens. With the focal length $f_{col}$, the core diameter $D_{core}$ defines the divergence angle $\theta_{div}$ and the aperture angle $\theta_{NA}$ defines the diameter $D_{spot}$ of the light spot on the sample. The mirror is rotatable and movable. The angle of light incidence on the sample is $\alpha_{slide}$. (For more details, see Methods.) (D) Measurements of light intensities of translucent light in the foveolar centre entering at different angles are shown. The mean loss of intensity + standard deviation at an angle of 5 and 10° was calculated as a percentage of the 0°'s value and was 15,3% + 9,6% and 67,8% + 11,3% respectively. (E) A human foveolar center is shown in translucent light entering at 0, 5 and 10° respectively. These images correspond to the measurements in (D). (F) Hypothetical explanation of the SCE-like drop of light intensity: light hitting the Müller cell at an angle is partly reflected (black dashed line) at the surface when entering the watery cytoplasm and reduces the transmission of light into the cones. Light not entering at an angle is fully transmitted into the cones (red line).

the transmitted light intensity is about 96% of the incident intensity, depending on the refractive index of the slide when $n_{slide} = 1.5255$ ($\lambda = 546$ nm, borosilicate glass/Schott). The intensity declines by about 4% when light incidence is under $\alpha_{slide} = 20°$. The angle $\alpha_{müller}$ of a light beam to a Müller cell can be distinguished from the angle of incidence on the slide through the multiple refractions of previous optical layers (glass material etc.). This angle is determined by the refractive index $n_{front}$ of the substrate before the Müller cell and the angle of incidence $\alpha_{müller} = \alpha_{slide}/n_{front}$.

The maximum acceptance angle (half angle) under which a light beam can penetrate the cell is

$$\Theta_{max} = \sin^{-1}\left(\frac{1}{n_{front}}\sqrt{n_{core}^2 - n_{gladding}^2}\right), \tag{1}$$

with the refractive indices $n_{core}$ of the substrate within the cell and $n_{gladding} < n_{core}$ of the cell edge.

The acceptance angle of the Müller cells *in vivo* is estimated with the refractive indices according to *Franze et al. (2007)* whereby $n_{front} = 1.358$ (neuron), when the light coupling of a cell does not connect to the vitreous body of the eye, and $n_{core} = 1.359$ (end foot). The refractive index of the cell edge is not known and is set as $n_{gladding} = n_{front}$. According to Eq. (1), the acceptance angle *in vivo* is $\theta_{max} = 2,2°$. When fixing the sample with phosphate buffered saline (PBS—buffer, $n_{PBS} = 1.33$) it can be assumed that the substrate surrounding the Müller cells has the index value $n_{gladding} = n_{PBS}$. The acceptance angle is then $\theta_{max} = 12,0°$ and is measured in the experiment with $\alpha_{slide} = n_{front} \cdot \theta_{max} = 16°$.

## Quantification of angle dependent foveal light transmission

Light micrographs with 100-fold magnification from the foveae and parafoveae of the three donors were taken under equal conditions with collimated light hitting the sample at a 0°, 5°, and 10° angle. The photos were processed using ImageJ 1.48v. The mean pixel intensity of the fovea or parafovea area was measured. The value of the image taken at the angle of 0° was set as 100% for each fovea. The mean loss of intensity $\pm$ standard deviation at an angle of 5° and 10° was calculated as a percentage of the 0°'s value.

## RESULTS

The first interesting finding was that the arrangement of disk membranes is more regular in rods compared to cones (Fig. 1A).

In semi- and ultrathin cross sections of fixed monkey foveolae, inner and outer segments of the cone photoreceptors were arranged in a highly ordered pattern (Fig. 1B). In the same plane of section, the inner segments were hit longitudinally, diagonally, and transversely in subsequent underlying rows (Figs. 1C and 4). These results indicate that the inner segments of the cones in the central foveola area are curved. We detected curved inner segments in 18 out of 21 monkey foveae. There were substantial differences in the shape of individual foveae, which corresponds to the variance among individuals observed in psychophysical measurements (*He, Marcos & Burns, 1999*). The diameter of the area containing curved inner segments was 267 µm $\pm$ 132 µm A calculation of cone numbers in the central part

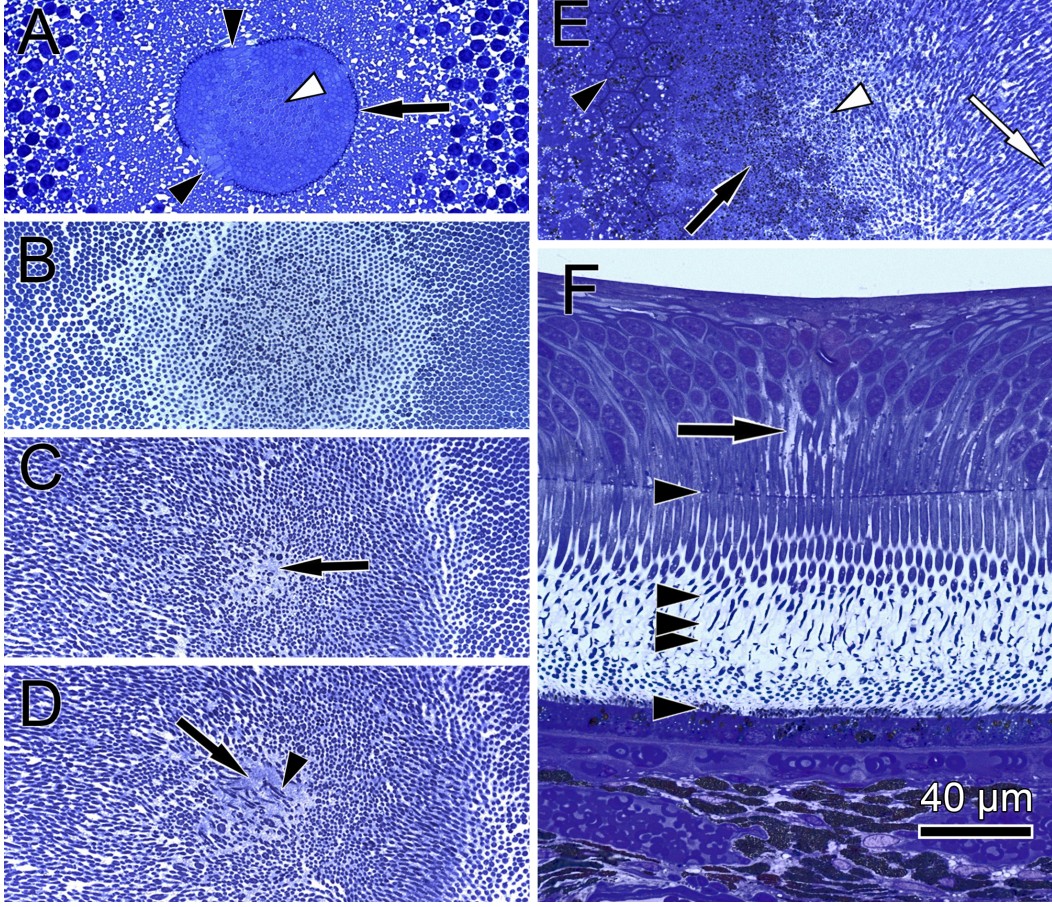

**Figure 4** **Distribution of central cone orientation in serial sections of the monkey fovea.** (A) At the level of the outer limiting membrane (arrow) inner segments are hit in different directions (arrowheads) in the same plane of section. (B) Cones are thin and separated. (C) The foveolar center is kept open (arrow). (D) Cone outer segments run perpendicularly (arrowhead) within the open center (arrow). (E) The tips of central cones (white arrowhead) reach the RPE (black arrowhead). Close to the RPE, cone outer segments run parallel (white arrow). The black arrow points to melanosomes of the RPE. (F) Arrowheads show the planes of section in (A–E). The arrow indicates central Müller cells.

of the foveola with a diameter of 200 μm results in approximately 6,500 cells. Of these cones, 75% of the inner segments are curved. These cones are located in the ring—like area between the central part of the foveola with a diameter of 100 μm. The remaining 25% of cone inner segments within the area of the central 100 μm diameter are straight. The pattern formed by the outer segments was of an even higher complexity but they were not straight in the central foveola (diameter of 200 μm).

The precise form of foveal fundus cells, however, can only be judged by three-dimensional (3D) reconstruction. To unravel the form and shape of foveal cells (Fig. 1D), we performed serial horizontal and vertical semi-thin sections through monkey foveae and constructed 3D models of foveal cones. The outer segments of cones within the central foveolae were indeed curved or even spiraly twisted (Fig. 1E and Video S2). In addition,

the central foveolar cone outer segments (52 µm ± 5.8 µm) were twice as long as the cone outer segments in the parafovea (26 µm ± 1,6 µm) (Fig. 1F).

To understand the three-dimensional shape of the central Müller cells, we formed 3D models of human and monkey retinae (Fig. 2D, Videos S3 and S4) from light and electron microscopic serial sections. The cross-section of individual Müller cells was often triangular (Video S5, bottom left). Focused ion beam (FIB) analysis shows that the Müller cells adapt to the shape of the cone nuclei and cross the retina in a wavy manner (Video S4). We found that individual foveolar Müller cells have a plateau zone at the site where their processes bend horizontally (Figs. 2C and 3B). Our model resembled the one described earlier (*Bringmann et al., 2006*).

We found that when the light enters the fovea of human retinae at an angle of 0° , the transmitted light forms a very bright spot in the center of the foveola (approximately 200 µm in diameter) (Fig. 3C). This area corresponds exactly to the area composed only of cones and Müller cells (*Bodis-Wollner, Glazman & Yerram, 2013*). However, when the angle of the transmitted light was changed to 10° , the bright spot in the foveolar center became dark (Fig. 3D, bottom right) and the SCE-like drop in light intensity became directly visible (Video S6). Measurements of the intensities of light transmission in the central foveola for incident angles of 0, 5, and 10° (Fig. 3F) resemble the relative luminance efficiency for narrow light bundles as a function of the location where the beam enters the pupil as reported by *Stiles & Crawford (1933)*. The effect was observed in all human foveae and persisted after carefully brushing away the outer segments. The 3D structure of human fovelolar Müller cells is shown in Video S7.

## DISCUSSION

Already in 1907 the unique central cones were described and called the ''bouquet of central cones'' (*Rochon-Duvigneaud, 1907*). The fact that the form and outer segment lengths of the central cones differ from that of the peripheral ones also was described in *Greeff & Graefe, (1900)* but their spatial organization remained unclear. In the present study we performed 3D modeling of central- and para-foveolar cones to investigate their exact anatomical shape, orientation and spatial arrangement.

The lengths of inner and outer segments and numbers of cones in our study correspond to reported measurements from monkey and human cones (*Borwein et al., 1980*; *Packer, Hendrickson & Curcio, 1989*; *Yuodelis & Hendrickson, 1986*). The 3D model of the central foveolar cones shows that outer segments do not run parallel to the incident light as reported earlier (*Hogan, Alvarado & Weddell, 1971*; *Laties, 1969*) but are curved or even coiled (Video S2) and proceed collaterally to the retinal pigment epithelium (RPE) (Fig. 1E). Our serial sections through the inner and outer segment layer of foveal cones clearly show that in the foveal center (100 µm) cones have unique shapes, directionalities, and distances from each other (Fig. 4). However, using these findings no convincing hypothesis for the origin of the SCE I could be proposed.

Thus, we searched for the origin of the SCE within the neural retina and performed sagittal and transverse serial sections through the foveolae of humans and monkeys to

construct 3D models and investigate its three-dimensional structure. Unexpectedly, we found extremely large Müller cells in the central foveola of monkeys (Fig. 2A and Video S1) and humans (Fig. 3A, Videos S7 and S4) in which cell organelles are rarely present (Figs. 2B, 2C and Video S5). These Müller cells were already described 47 years ago (*Yamada, 1969*) in human eyes as having an unusual watery cytoplasm but have been neglected (*Gass, 1999*).

As light propagation by Müller cells through the retina (*Franze et al., 2007*) has been shown to be important and increases photon absorption specifically by cones (*Labin et al., 2014*), we hypothesize that light hitting the Müller cell plateau at an angle of 0° is effectively transmitted into the photoreceptors whereas light hitting the Müller cell plateau at a different angle is partly reflected accordingly, which reduces the amount of light guided through the Müller cells (Fig. 3F). This is in accordance with the finding that retinal foveal structures reflect light entering at an angle different from 0° (*Gao et al., 2008*; *Van de Kraats & Van Norren, 2008*).

To test this hypothesis, we used human foveae from flat mounted isolated retinae and measured the transmission of collimated light under the light microscope at different angles. We could indeed measure a SCE-like decrease in the transmitted light intensity when the angle of the lightbeam was deflected from 0° . We do not completely exclude that the Henle fibres and the shape of the foveal pit are also involved in light reflection. It is also not ruled out that the specific shape of the cone outer segments plays an additional role in causing the SCE by possible angle dependent sensitivity of conopsins for photons.

The spatial resolution of human vision decreases by 50% when a subject in the view is deviated 2.3° from the foveation point (*Zhang et al., 2010*). The present findings may be physiologically involved in foveation, which is a process of bringing eccentric targets to the direct sight line by saccadic eye movements. This process may be accelerated by reflection of photons entering at an angle different from the direct sight line.

This study might also add a new piece to the puzzle of the pathogenesis of macular telangiectasia type 2 which is characterized by a loss of Müller cells and a reduction of SCE (*Powner et al., 2013*; *Zhao et al., 2015*).

## CONCLUSIONS

This paper shows the 3D anatomy of primate foveolar Müller cells and cones for the first time. Outer and inner segments of foveolar cones in monkey eyes have directionality and are not arranged in a perfectly straight manner in the axis of the straight entering light. Unique Müller cells with optical fibre characteristics are present in the center of the foveola. These findings may be involved in the foveation process.

In addition, our findings could be of interest for the understanding of the pathogenesis of macular telangiectasia type 2. Finally a new hypothesis for foveal Müller cells, causing the SCE by angle-dependent light reflection, is presented.

## ACKNOWLEDGEMENTS

We thank Prof. Susanne Trauzettel-Klosinski and Prof. Brian Vohnsen for constructive discussion, Hanna Janicki and Sigrid Schultheiss for technical support, Dr. Birgit Schröppel for FIB analysis and Judith Birch for editorial assistance.

### Funding

We received support from the Deutsche Forschungsgemeinschaft and the Open Access Publishing Fund of the University of Tübingen. The funders had no role in study design, data collection and analysis, decision to publish, or preparation of the manuscript.

### Grant Disclosures

The following grant information was disclosed by the authors:
Deutsche Forschungsgemeinschaft.
Open Access Publishing Fund of the University of Tübingen.

### Competing Interests

The authors declare there are no competing interests.

### Author Contributions

- Alexander V. Tschulakow and Ulrich Schraermeyer conceived and designed the experiments, performed the experiments, analyzed the data, contributed reagents/materials/analysis tools, prepared figures and/or tables, authored or reviewed drafts of the paper, approved the final draft.
- Theo Oltrup performed the experiments, analyzed the data, contributed reagents/materials/analysis tools, prepared figures and/or tables, authored or reviewed drafts of the paper, approved the final draft.
- Thomas Bende analyzed the data, authored or reviewed drafts of the paper, approved the final draft.
- Sebastian Schmelzle contributed reagents/materials/analysis tools, prepared figures and/or tables, authored or reviewed drafts of the paper, approved the final draft.

### Human Ethics

The following information was supplied relating to ethical approvals (i.e., approving body and any reference numbers):

The human eyes were gifts from the Clinical Anatomy of the University of Tübingen (ethical number for scientific issues 237/2007B01).

### Animal Ethics

The following information was supplied relating to ethical approvals (i.e., approving body and any reference numbers):

Monkeys were kept at Covance Laboratories GmbH (Münster, Germany study numbers 0382055, 8260977, 8274007) or SILABE-ADUEIS (Niederhausbergen, France). The

Covance Laboratories GmbH test facility is fully accredited by the Association for Assessment and Accreditation of Laboratory Animal Care (AAALAC). This study was approved by the local Institutional Animal Care and Use Committee (IACUC), headed by Dr. Jörg Luft and the work was carried out in accordance with the Code of Ethics of the World Medical Association (Declaration of Helsinki). The monkeys from SILABE-ADUEIS were euthanized due to veterinarian reasons. Since they had not been included in a study before, they do not have a study number.

## Data Availability

Figshare

https://doi.org/10.6084/m9.figshare.5769507

https://doi.org/10.6084/m9.figshare.5769501

https://doi.org/10.6084/m9.figshare.5769444

https://doi.org/10.6084/m9.figshare.5769447

https://doi.org/10.6084/m9.figshare.5769450

https://doi.org/10.6084/m9.figshare.5769420

https://doi.org/10.6084/m9.figshare.5769399

https://doi.org/10.6084/m9.figshare.5769372

https://doi.org/10.6084/m9.figshare.5769366.

## Supplemental Information

Supplemental information for this article can be found online at http://dx.doi.org/10.7717/peerj.4482#supplemental-information.

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
