# Peer review of "The anatomy of the foveola reinvestigated"

_PeerJ, doi:10.7717/peerj.4482_

## Round 0.1 · original submission · Minor Revisions

Please consider, and answer, all the reviewers' comments.

Reviewer 1 ·

Basic reporting

The background is clear and sufficient.
The aim is relevant and according with the procedures.

Experimental design

Research question is clear and important.
methods are well presented, and are of high quality.
Ethical aspects are well explained.

Validity of the findings

I Consider they showed interesting and meaningful results. But presentation can be improved.

Additional comments

The authors analyzed the structure of primate foveola and correlate it with the Stiles Crawford Effect of the first kind (SCE). They followed electron microscopic studies as well as ion beam/scanning electron microscopy and shown elegant 3D images. The results support the hypothesis that the anatomy of the foveola result in the SCE effect.
The field background is good. The results are interesting and propose that Müller cells might participate in the SCE.
However, I consider some points should be taken in account previous to be accepted for publication.
1. All the results should be presented in the corresponding section; then, lines 290-305 (discussion) should be at the result section. Following that, the figures (supplemental and videos) must have a sequential order (1-7, not 1,4,6…). Also, suppl. figure 1 represent the principal anatomic results, so should not be supplementary.
2. Please check the figures, I could not find panels E and F in Fig. 2.
3. Legend of Fig.3 should be modified according with the panels and each one explained, indicating the meaning of the different symbols.
4. Line 290, the measurements obtained in this work must be presented in the result section.
5. The size of the figures, particularly the S1, is not sufficient to identified the type of cells indicated. There is possible to identified Müller cells with an specific antibody?
6. Since the cones were found to be curved, that can also participate in the SCE, please discuss.

Minor comments:
1. Line 118, indicate type of fixative.
2. Line 271, reference is not complete.

·

Basic reporting

The paper is clear and unambigous, all the Literature references needed are included, with professional structure, but some specific flaws in figures and the list of references (please see notes to the authors); the author share raw data

Experimental design

This research is original, the authors define very well the research purpose which is relevant and meaningful; methods are described even though they need more details in specific points, i.e. in: Material and Methods, line 167
For Fig. 3d and Video 3, the areas of interest were selected automatically by adjusting the threshold.

This statement is too much colloquial, Threshold of what? Which parameter was taken into account?

Otherwise, the investigation was conducted with the highest standards

Validity of the findings

Even though the image processing is quite robust and impressive by its beauty, I do have some concerns about the overall analysis of data, for instance in

Results, line 254
We detected curved inner segments in 18 out of 21 monkey foveae. There were substantial differences in the shape of individual foveae, which corresponds to the variance among individuals observed in psychophysical measurements(He et al. 1999). The diameter of the area containing curved inner segments was 267 μm + 132 μm. The pattern formed by the outer segments was of an even higher complexity.

The authors talk about the area, but they do not deal anything on amount of cells either as an absolute number or percentage or any other ratio of curved cells compared to “regular” cells. It must be assumed that all the cells showed the reported morphology? If not, what about to perform an statistical analysis of data.

Another example is:

Results, line 263

In addition, the central foveolar cone outer segments (52 μm) were twice as long as the cone outer segments in the parafovea (26 μm).

Is there any numerical assessment for this assertion, i.e how many long outer segments out of 100 or whichever amount of cone outer segments?

Also, the conclusions are well stated, but it looks to me that in some points is too much speculative, specifically in:

Discussion, line 306
As light propagation by Müller cells through the retina (Franze et al. 2007) has been shown to be important and increases photon absorption specifically by cones (Labin et al. 2014), we hypothesize that light hitting the Müller cell plateau at an angle of 0 degrees is effectively transmitted into the photoreceptors whereas light hitting the Müller cell plateau at a different angle is partly reflected accordingly, which reduces the amount of light guided through the Müller cells. This is in accordance with the finding that retinal foveal structures reflect light entering at an angle different from 0 degrees.

On Figure 2, panel B the authors describe the Müller cells in this area as lacking organelles, Unique central foveolar Müller cells from monkeys (A) In a semi-thin section perpendicular to the optical axis the central Müller cells are translucent (arrow). (B) The Müller cells (arrow) are shown in the same orientation as in a) at high magnification. They do not contain cell organelles.
So, how could thus reflect light, when there is no refringent material inside them? Please elaborate.

Additional comments

My general comments refer on typos, mainly:


Introduction, line 67:
Stiles also showed that monochromatic light of the same wavelength
It looks redundant to me, could rephrase to “Stiles also showed that monochromatic light”.

Material and Methods, line 121
This protocol has been shown to minimize fixation artefacts.


Is there in literature reference for that statement or comes form Group experience?. Please explain it.


Figure 1
Anatomical findings in monkey foveae and cones
(A) In an electron micrograph, an extra foveal cone contains irregularly ordered stacks of photoreceptor disc membranes (black arrowheads) and spaces free of photoreceptor membranes (arrow). In contrast, the disk membranes in rods are highly ordered (white arrowhead).

There is no white arrowhead in panel (A)

Figure 2
Unique central foveolar Müller cells from monkeys (A) In a semi-thin section perpendicular to the optical axis the central Müller cells are translucent (arrow). (B) The Müller cells (arrow) are shown in the same orientation as in a) at high magnification. They do not contain cell organelles. The arrowhead marks a Henle fibre. (C) The plateau zone of a Müller cell is indicated by a white arrowhead in a section parallel to the optic axis. Henle fibers are indicated by a black arrowhead and cone nuclei by (N) (D) A 3D model shows the main part of the central Müller cells of a monkey. ( E) The central Müller cell in Fig. b (arrowheads) is shown integrated into a stack mounted from serial sections. (F) A human foveolar Müller cell 3D model is shown and its plateau zone marked by an arrow.
<!--[if !supportLineBreakNewLine]--> <!--[endif]-->

There are no panels E nor F in the figure

References, line 348
Brinkmann;A RA. Müller cells in health and disease

There are missing bibliographical data

---

## Round 0.2 · accepted · Accept

Congratulations! Your article has been accepted for publication in PeerJ.

Reviewer 1 ·

Basic reporting

No comment.

Experimental design

No comment.

Validity of the findings

No comment.

Additional comments

The authors did most of the suggested corrections.

·

Basic reporting

I received the second version of the manuscript, which includes all the modifications according to my first review. The manuscript is modified accordingly and my suggestiuon is tobe accepted as is

Experimental design

I received the second version of the manuscript, which includes all the modifications according to my first review. The manuscript is modified accordingly and my suggestiuon is tobe accepted as is

Validity of the findings

I received the second version of the manuscript, which includes all the modifications according to my first review. The manuscript is modified accordingly and my suggestiuon is tobe accepted as is

Additional comments

I received the second version of the manuscript, which includes all the modifications according to my first review. The manuscript is modified accordingly and my suggestiuon is tobe accepted as is